# Potential Role of Omega-3 Polyunsaturated Fatty Acids in Pediatric Food Allergy

**DOI:** 10.3390/nu14010152

**Published:** 2021-12-29

**Authors:** Marco Ugo Andrea Sartorio, Erica Pendezza, Serena Coppola, Lorella Paparo, Enza D’Auria, Gian Vincenzo Zuccotti, Roberto Berni Canani

**Affiliations:** 1Department of Pediatrics, Vittore Buzzi Children’s Hospital, University of Milan, 20154 Milan, Italy; marcoua.sartorio@gmail.com (M.U.A.S.); erica.pendezza@unimi.it (E.P.); gianvincenzo.zuccotti@unimi.it (G.V.Z.); 2Department of Translational Medical Science—Pediatric Section, University of Naples “Federico II”, 80131 Naples, Italy; sery.cop28@gmail.com (S.C.); paparolorella@gmail.com (L.P.); berni@unina.it (R.B.C.); 3ImmunoNutrition Lab at CEINGE-Advanced Biotechnologies, University of Naples “Federico II”, 80131 Naples, Italy; 4European Laboratory for the Investigation of Food-Induced Diseases, University of Naples “Federico II”, 80131 Naples, Italy; 5Task Force on Microbiome Studies, University of Naples “Federico II”, 80131 Naples, Italy

**Keywords:** omega-3 polyunsaturated fatty acids, food allergy, immune system, allergy prevention, nutritional sciences, pediatrics

## Abstract

Polyunsaturated fatty acids (PUFAs) are involved both in immune system regulation and inflammation. In particular, within the PUFAs category, omega-3 (ω-3) may reduce inflammation, whereas omega-6 (ω-6) PUFAs are generally considered to have a proinflammatory effect. Recent evidence highlights an imbalance in the ω-3:ω-6 ratio with an increased intake of ω-6, as a consequence of the shift towards a westernized diet. In critical age groups such as infants, toddlers and young children, as well as pregnant and lactating women or fish allergic patients, ω-3 intake may be inadequate. This review aims to discuss the potential beneficial effects of PUFAs on pediatric food allergy prevention and treatment, both at prenatal and postnatal ages. Data from preclinical studies with PUFAs supplementation show encouraging effects in suppressing allergic response. Clinical studies results are still conflicting about the best timing and dosages of supplementation and which individuals are most likely to benefit; therefore, it is still not possible to draw firm conclusions. With regard to food-allergic children, it is still debated whether PUFAs could slow disease progression or not, since consistent data are lacking. In conclusion, more data on the effects of ω-3 PUFAs supplementation alone or in combination with other nutrients are warranted, both in the general and food allergic population.

## 1. Introduction

Polyunsaturated fatty acids (PUFAs) are a group of fatty acids with a molecular structure characterized by more than one double bond.

The two main families of PUFAs are the omega-3 (ω-3) and omega-6 (ω-6) families, which differ in the position of the first double bond [1]. The precursors of these families are linolenic acid (ALA; 18:3ω-3) and linoleic acid (LA; 18:2ω-6), respectively [2]. Linolenic acid and LA are defined as essential fatty acids since they cannot be synthesized by humans, so they must be obtained necessarily from the diet [2]. Evidence suggests that PUFAs may contribute both to immune system regulation and allergy development. PUFAs are involved, for instance, into cell membrane fluidity modification, production of inflammatory and anti-inflammatory mediators, and can also affect gene expression. Within the PUFAs category, ω-3 may reduce inflammation and improve allergic symptoms [3], whereas ω-6 PUFAs are generally considered to have a proinflammatory effect since they favour T helper cell 2 (Th-2) immune response and allergy development [4].

Over the last decades it has been observed that dietary changes, with a shift towards a “westernized” diet, have led to a general imbalance between the amount of ω-3 and ω-6 consumed in favour of ω-6 PUFAs [1]. The western diet is characterized by a high dietary ratio of ω-6:ω-3 PUFAs, which are derived from increased fat and vegetable oils (including soybean, corn, sunflower, safflower oil, cotton seed oils) containing proinflammatory ω-6 PUFAs, and reduced fiber, potentially contributing to allergic sensitization and to severity of allergic symptoms [5,6,7]. While the typical Western diet has a much greater ratio of ω-6 PUFAs compared with ω-3 PUFAs, research has shown that increasing the ratio of ω-3 to ω-6 fatty acids in the diet may decrease the incidence of many immune-mediated chronic conditions, including food allergy (FA).

Allergic diseases in infancy may present as a continuum where they progress from one to another in the so-called “atopic march” [8]. Classically, the atopic march begins with atopic dermatitis and then progresses to food allergy and respiratory allergy, including rhinitis and asthma. It is known that atopic dermatitis correlates with the risk of developing FA [9]. This could be due to the transcutaneous sensitization to food allergens via an inflamed and disrupted skin, according to the hypothesis of dual-allergen exposure, which suggests that allergic sensitization to food occurs through low-dose cutaneous sensitization, whereas early consumption of food protein induces oral tolerance [10].

Thus, the ability of ω-3 PUFAs to prevent or reduce skin inflammation might also exert a protective role on food allergy cutaneous sensitization [11], contrasting the progression of the atopic march. To date, literature about the potential preventive role of ω-3 PUFAs supplementation on atopic diseases is still scarce, and existing trials mostly focused on atopic dermatitis or wheezing rather than food allergy, an issue which needs to be further investigated.

The present review aims to focus on the potential role of ω -3 polyunsaturated fatty acids in food allergy prevention and treatment; however, as many studies considered multiple allergic outcomes, including food allergy development, these studies have been included and are discussed. Knowledge gaps are also addressed. A comprehensive search was conducted using the electronic databases MEDLINE via PubMed (www.pubmed.gov, accessed on 19 July 2021) and Embase databases (www.embase.com, accessed on 19 July 2021).

The following keywords were used: ω-3 polyunsaturated fatty acids, ω-6 polyunsaturated fatty acids, polyunsaturated fatty acids, eicosapentaenoic acid, docosahexaenoic acid, food allergy, immune system, allergy prevention, allergy treatment, children, pediatric food allergy, pregnancy PUFAs supplementation, lactation PUFAs supplementation, children PUFAs supplementation, human milk.

## 2. ω-3 PUFAs: Sources, Metabolism, and Recommended Intake

The main sources of ALA are linseeds, chia seeds, vegetable oils such as linseed, soybean, rapeseed and wheat germ oils, and walnuts [2,12]. Linoleic acid is found mostly in vegetable oils such as soybean, sunflower, corn seed, wheat germ and sesame oils [2,12]. Sources of arachidonic acid (AA) are meat and meat products, egg yolks and offal [2]. Eicosapentaenoic acid (EPA) and docosahexaenoic acid (DHA) are present in most fish and seafood products, and especially in oily fish and fish oil (Table 1).

After ingestion, ALA and LA are metabolized in the liver by desaturases (5 and 6) and elongases to obtain long chain PUFAs: EPA, docosapentaenoic acid (DPA) and DHA from ALA in the ω-3 group, and AA from LA in the ω-6 group [1,14]. Since the ω-3 and ω-6 metabolic pathways share the same elongation and desaturation enzymes, they compete for conversion [1]. PUFAs also act as precursors to several bioactive metabolites, such as eicosanoids, including prostaglandins, leukotrienes and thromboxanes [14].

In Table 2 the Dietary reference values (DRVs) for PUFAs provided by the European Food Safety Authority (EFSA)’s Panel on Dietetic Products, Nutrition and Allergies are reported [15].

Data regarding PUFAs intake are rather far from recommendations. A systematic review recently assessed the intake of ω-3 and ω-6 PUFAs in specific population groups, including pregnant and lactating women, infants (aged 6 to 12 months) and children, from 17 different European countries. Results were compared to the EFSA recommendations. The results indicated that intakes of ω-3 and ω-6 PUFAs were largely suboptimal in specific population groups in Europe [16]. This is especially significant in critical age groups that have a higher requirement for these fatty acids and are therefore exposed to an increased risk of inadequate intake; among these groups are infants, toddlers, and young children, as well as pregnant and lactating women.

## 3. ω-3 PUFAs in Human Milk

The composition of human milk differs among lactating women. These differences can be particularly relevant for the concentration of fatty acids [17]. ALA and LA are the main PUFAs present in human milk. Analysing human milk PUFAs content in nine European countries, a content of 0.9% wt/wt for ALA and 11.0% wt/wt for LA [18] was found.

A meta-analysis reviewing 65 studies reporting data from 2400 women worldwide indicated a mean concentration of DHA in human milk of 0.32 ± 0.22% of total fatty acids, with a range from 0.06 to 1.4%. Higher DHA concentrations were found in populations living near the sea and were associated with the consumption of fish and seafood [19]. A recent review by the European Academy of Allergy and Clinical Immunology (EAACI) addressing food allergy prevention suggested that breastfeeding may not exert a protective role on FA development; however, the evidence is small since only seven observational studies were included in the revision. Despite this, since human milk provides all of the nutritional needs of infants up to six months of age, EAACI recommends healthcare professionals to encourage breastfeeding [20,21].

## 4. PUFAs Dietary Intake in Healthy and Allergic Children

Proper timing for the introduction of fish, the main source of PUFAs, in infant diets remains debated: early introduction (before 9 months of life) may increase diet diversity and have potential beneficial effects. An increased diet variety during the first year of life may provide an increased amount of healthy nutrients and may also play a protective role in allergy development. Allergy prevention may be linked to early exposure to food allergens during immune system development but can also be related to early exposure to those nutrients which exert an immune modulating effect and prevent allergic sensitization. It has been demonstrated that the introduction of each additional food at 6 months of age reduced the risk of developing FA over the first 10 years of life [22]. These results suggest that diet diversity is associated with increased nutrient intake, including those nutrients which could have a protective role in allergy development, in particular ω-3 fatty acids and nondigestible fibers [23,24].

The American Academy of Paediatric Guidelines on timing of introduction of allergenic complementary foods states that delaying the introduction of allergenic foods such as fish beyond 4 to 6 months does not prevent atopic disease [25]. As fish represent the main source of ω-3, introduction is recommended during the second semester of life with a variability depending on child development, family believers and socio-cultural factors [26].

Aldámiz-Echevarría et al. showed that food-allergic children on an elimination diet are at risk of developing ω-3 PUFAs deficiency [27]. Since fish are the most relevant PUFAs source, allergy to fish may obviously lead to inadequate intake of ω-3 PUFAs in children [26]. However, being allergic to one fish does not necessarily mean being allergic to all fish species [28]. For this reason, if the clinical history and/or the presence of specific IgE (sIgE) to one or more fish suggest the suspicious of FA, it is mandatory to perform an oral food challenge to confirm or to exclude an allergic reaction. Children who are allowed to consume at least two portions of fish per week reach an adequate amount of EPA and DHA [13].

In contrast, children who are not permitted to eat any type of fish should be supplemented with algal-derived ω-3 fatty acids [29,30].

Similarly, supplementation is recommended in children who follow a vegetarian or vegan diet, as this dietary pattern implies the exclusion of animal-derived proteins, including fish, resulting in a reduced or null ω-3 intake.

Noteworthy is that a recent trial (2020) by Ogrodowczyk and colleagues investigated the associations between maternal dietary pattern and immunological markers (such as specific IgE profile and cytokines) in the offspring. A vegetarian/vegan diet was characterized by “prevalence of vegetables, fruits, juices, dark rice, and meat substitutes based on pulses, nuts, grains and herbs”, together with a consistent elimination of animal products from the diet. The authors found that this dietary pattern seemed to be associated with high levels of total and specific IgE to several foods (cows’ milk proteins, eggs, soybean, nuts, beef) and higher levels of interleukins in the offspring. One possible explanation is that early exposure to phytoestrogens can increase IgE production later. 

However, as the same authors concluded, this association might be due to multiple confounding factors such as infant feeding type and birth method; therefore, additional studies are needed to draw firm conclusions [31].

## 5. Role of ω-3 PUFAs in the Regulation of Immune System Development and Function

A large subset of cells and cytokines are involved in immune system development and function. Among them, CD4+ helper T (Th) cells are immune cells able to differentiate into different subgroups including Th1 and Th-2 cells, which play important roles in immunity by different patterns of cytokines secretion. Th1 cells stimulate the cellular immune response and favor IgM and IgG1 production by B cells, while Th-2 cells through cytokines production (IL4, IL5, IL13) are associated with IgE production and eosinophilic and mast cell recruitment resulting in allergic inflammation. The balance between ω-3 and ω-6 PUFAs is involved in the subject’s propensity to develop allergic inflammation [32]. Th-2 inflammatory response promotes eosinophil chemotaxis and activation and IgE production by B lymphocytes.

During the period of immune programming, diet, environmental factors, microbiome and epigenetic mechanisms may play an important role in the development of FA [33].

## 6. PUFAs Supplementation for Food Allergy Prevention: Preclinical Findings

Several preclinical studies have indicated ω-3 PUFAs as protective molecules against allergic inflammation [34,35,36,37,38]. In a murine model of allergy, DHA inhalation during the allergen challenge phase in mice suppressed airway eosinophilic inflammation, and this was accompanied by reduced numbers of inflammatory cells in bronchoalveolar lavage fluid and decreased airway hyperresponsiveness, and mucus production [38]. The new monoglyceride DHA derivative (CRBM-0244) and EPA derivative (EPA-MAG) showed preventive effects on airway eosinophilic inflammation, airway hyperresponsiveness and inflammatory cytokine production in OVA-induced asthmatic responses [36,37]. In mice with orally induced hen’s egg allergy, feeding with 13% salmon oil diet containing 6.1% EPA and 7.5% DHA partially prevented development of FA [39], although no differences in immunoglobulin levels were observed. Other studies in an animal model of FA reported that DHA and EPA were able to prevent and reduce cow’s milk and peanut allergies [40] by reducing IgE, IgG1, and IgG2a levels and increasing Treg number, while lowering both Th-2 and Th1 activation [41].

An increased dietary ω-6/ω-3 PUFAs ratio may induce a shift in the Th1/Th-2 balance towards a Th-2 response [42]. It has been demonstrated that the ratio of IFN-γ/IL-4 production by human peripheral blood mononuclear cells was significantly higher during consumption of ω-3 PUFAs compared to a ω-6 PUFAs-rich diet, as a result of reduced IL-4 secretion [42]. The protective effects of DHA against allergic disease is due, at least in part, to its capacity to reduce IgE production by human B cells [43], to suppress Th-2 polarizing and to enhance IL-10 production, with an increase of dendritic cell (DC) number [42,44].

The protective effect of ω-3 PUFAs during the allergic sensitization phase is not only limited to their ability to antagonize antigen presentation to naive T-cells but also to a reduction of DC immunogenicity. In vitro studies by Zeyda et al. demonstrated that PUFAs administration to monocyte-derived DCs may block immunogenic function, causing an altered DC surface molecule expression and a reduction of cytokine release [45]. Similar results from Kong et al. showed that exposure of bone marrow-derived DC to DHA resulted in prolongation of immature phenotype and significant decrease of proinflammatory cytokine production and release [46]. Immature DC triggers a lower immune response together with a pattern of cytokine secretion, which induce a more tolerogenic state [47].

DHA and EPA modulate allergic inflammation by binding to several receptors, such as GPR120 and nuclear receptor PPARα/γ [48] and inhibit MHC-II expression, activation of CD86 through TLR4, expression of costimulatory molecules (CD40, CD80, and CD86) and inflammatory cytokine production (IL-6 and IL-12p70) in DCs [49]. Interestingly, T-cells cocultured with DHA-treated murine DCs expressed higher expression of transforming growth factor (TGF) β and forkhead box P3 (Foxp3) [46]. The stimulation of DCs with DHA significantly decreased the expression of DC surface marker OX40L, as well as the IL-12p40, IL-23 cytokine production and Th-2-type IL-13 response by DCs [50]. The effective DHA concentrations able to elicit these biological effects range from 2 to 100 μM [51,52,53,54].

PUFAs metabolism involves the desaturases enzymes, Δ-5 and Δ-6 desaturases, encoded by FADS1 and FADS2 genes, respectively, localized on chromosome 11q12-13.1 [55]. Previous studies have studied the correlation between the genetic variants in the FADS gene cluster and PUFAs serum levels [56,57,58,59,60,61,62]. Polymorphisms in FADS2 were found negatively associated with activity of Δ-6 desaturase and EPA production [63]. A study demonstrated the association between a reduced activity of Δ-6 desaturase, the presence of FADS2 polymorphisms and lower PUFAs serum levels in subjects at 13 years of age [64], although the evidence on correlation between the FADS polymorphisms and FA is still controversial [56,61,64]. Increasing evidence suggests that genetic variation in fatty acid desaturase (FADS) genes influence the biological PUFAs status and the effect of dietary intake on PUFAs concentrations [65]. Given the impact of FADS genotype on PUFAs concentrations, it is likely that the genotype has the potential to influence health outcomes. This issue has been investigated in relation to different health outcomes, including allergic diseases, in particular asthma and eczema, with no conclusive results [61,65,66]. To our best knowledge, no studies have specifically investigated the effect of genetic variations on development of food allergies. The role of genetic variation on child health outcomes, including food allergies, needs to be further investigated, taking into account FADS genotype of both mother and child, as well as maternal and child FAD status.

## 7. PUFAs Supplementation during Pregnancy and Breastfeeding for Allergy Prevention: Clinical Studies

The prenatal period is a sensitive time, during which intrauterine exposures can modulate the course of development and confer an enduring effect on the offspring [66]. Since immune system development in early life is not yet complete, it can be modulated from environmental factors such as maternal diet [67]. Despite the inconclusive and conflicting results, several studies and meta-analyses investigated the role of ω-3 PUFAs supplementation in pregnancy on offspring allergy [68,69,70,71,72,73,74]. Table 3 shows the clinical evidence on ω-3 PUFAs supplementation and the occurrence of FA or food allergens sensitization.

Maternal supplementation in the DOMinO trial with 900 mg of ω-3 PUFAs, starting from 21 weeks of gestation until delivery, did not protect offspring against food allergen sensitization or allergy, although egg sensitization was lower than offspring of the control group (received matched vegetable oil capsules without ω-3 PUFAs) [69,79]. Similarly, hen’s egg sensitization through the skin prick test was reduced by more than 50% in infants of women who received fish oil (3.7 g ω-3 PUFAs per day) compared to placebo in pregnancy from 20 weeks of gestation until delivery [75].

Unbalanced PUFAs levels have been associated with FA. Children affected by FA showed a decrease of plasma content in ω-3 PUFAs, and particularly EPA and DHA [22,69]. A recently published study showed that antenatal fish oil supplementation was associated with higher DHA and ω-3 related metabolites at 1 year of age and, conversely, a higher ω-6 PUFAs level was associated with higher odds of FA [80]. Instead, in a double-blind randomized controlled trial conducted in 420 infants at high atopic risk showed that daily fish oil supplementation (containing 280 mg DHA and 110 mg EPA) improved infant ω-3 status (higher levels of DHA and EPA) compared to the control (olive oil), but did not prevent childhood allergic disease [78]. Another multicenter, randomized controlled trial performed on a large number of preterm infants (n = 657) breastfed from mothers taking either tuna oil (high-DHA diet, ∼1% total fatty acids) or soy oil (standard-DHA, ∼0.3% total fatty acids) capsules, showed no preventive effect on FA at 18 months of life [76]. Despite that, the study had the limitation that the incidence of FA was stated by parental reporting.

The ω-3 PUFAs protective effect against FA in the offspring may further increase when supplementation is prolonged (e.g., extended during breastfeeding). Maternal supplementation with 2.7 g ω-3 PUFAs per day starting from week 25th of gestation until 3rd–4th month of breastfeeding in an at-risk population significantly lowered the FA prevalence in the offspring at 1 year of age [68]. Thus, it is clearly demonstrated that dosing, timing, and duration of maternal intervention contribute significantly to the FA preventive effect. In the same cohort of children, the ω-3 PUFAs preventive effect was still measured at the age of 2 years, and new cases of FA were not prevented. Furthermore, high proportions of EPA and DHA in maternal and infant plasma were associated with the lowest chance of FA development and the severity of other allergies was reduced [77].

Furthermore, several lines of evidence have demonstrated associations between maternal lipidomic profiles and offspring FA risk. It was recently shown that a high content of long chain fatty acids and double bond numbers predominantly composed of PUFAs in maternal blood triacylglycerols was associated with up to a 40% risk reduction of FA in the offspring [81]. Maternal PUFAs, especially ω-3 PUFAs, may regulate functions of the fetal immune system through several anti-inflammatory mechanisms and/or result in a reduced Th-2 response, thus protecting against FA [82,83,84].

In conclusion, the apparent absence of deleterious side effects of ω-3 PUFAs treatments is neither a conclusive demonstration of their efficacy, nor a suggestion for an indiscriminate supplementation. It is necessary to intensify scientific efforts to clarify their actual effects to align medical and nutritional guidelines.

It should also be noted that all the above-mentioned trials focused on PUFAs supplementation in the form of oil or fish oil capsules, with standardized dosages.

By contrast, PUFAs intake through diet (mainly depending on fish consumption) has great variability among different regions and populations, and among people living in the same geographical area but with different dietary habits; this could explain the high variability in the trials’ results.

## 8. PUFAs Supplementation in Infancy and Childhood for Allergy Prevention

The effect of ω-3 PUFAs supplementation in infancy and childhood in preventing allergic diseases has also been investigated, and results are also conflicting. This could be due to a lack of standardization in population identification (high vs. low risk), underlying diet (breastfeeding vs. standard infant formula vs. free diet), baseline PUFAs serum levels and dose, timing and duration of supplementation [85].

Fish oil supplementation in infants and children resulted in higher ω-3 PUFAs levels, but it is not clear whether these findings have clinical relevance [86].

While clinical intervention studies suggest a possible role of fish oil supplementation in infants/children in reducing wheezing and the need of bronchodilators, a meta-analysis [87] that included five relevant pediatric RCTs did not show significant reduction of asthma incidence with fish oil supplementation. This meta-analysis included a total of 2415 children receiving ω-3 PUFAs dietary supplementation with fish oil capsules administered by mouth for an average of 0–12 months and up to five years with an average duration of follow-up of 3–5 years, up to a maximum of 8 years. Among studies reported in the meta-analysis, the CAPS (Childhood Asthma Prevention Study) study interestingly showed in the active diet group receiving ω-3 PUFAs supplementation an increase in ω-3/ω-6 ratio at follow up, but this result did not match with any clinical effect or beneficial effect on lung function [88]. Conversely, Birch et al. [89] showed that term infants who had received PUFAs added to formula had a lower risk of wheezing, wheezing plus atopic dermatitis and any allergy at 3 years follow-up.

Foiles et al. [90] highlighted that infants receiving infant formula plus PUFAs for the first year of life had less allergic symptoms than those who received standard formula until 4 years old. PUFAs also lowered the prevalence of asthma in children at risk for atopy.

A double-blind randomized controlled trial by D’Vaz et al. [78] showed that ω-3 fatty acids supplementation via fish oil from birth to six months of age in infants at high risk of allergy may have a protective role against wheeze and eczema, but no differences were reported in term of sensitization and FA at 6 months. At a 1 year follow up, no protective effect on clinical outcomes was noted [78].

Interestingly, an Iceland cohort study showed that ω-3 PUFAs supplementation in infancy may decrease food sensitization and FA, although the latter was without statistical significance, and reduced severity of allergic symptoms [91].

The potential role of supplementation with PUFAs in FA children has not been investigated yet but could be of great interest to examine the immunomodulatory effect of ω-3 PUFAs supplementation on the allergic phenotype.

## 9. Conclusions

ω-3 PUFA may be particularly important for FA prevention and management. Discrepancies exist regarding the best timing and doses for ω-3 PUFA supplementation for prevention purposes, and which individuals are most likely to benefit, also taking into account the role of genetic variation, which needs to be further investigated. These limitations make it difficult to draw firm conclusions. Furthermore, most clinical studies have focused only on FA prevention. More data on the effects of ω-3 PUFA supplementation alone, or in combination with other nutrients with antioxidant and immunoregulatory properties, in FA patients are warranted.

## Figures and Tables

**Table 1 nutrients-14-00152-t001:** Eicosapentaenoic acid (EPA) and docosahexaenoic acid (DHA) content in different fish species [13].

EPA (g/100 g)	DHA (g/100 g)
Herring 1.09 g Salmon 0.89 g Tuna 0.80 g Grey mullet 0.76 g Mackerel 0.73 g	Tuna 2.15 g Grouper 1.45 g Mackerel 1.26 g Salmon 1.19 g Sardines 1.16 g Herring 1.01 g Swordfish 0.56 g

**Table 2 nutrients-14-00152-t002:** Recommended polyunsaturated fatty acids (PUFAs) daily dose.

		AI	RI
Infants (6–12 months)	PUFAs	-	
	ω-6 PUFAs	-	
	ω-3 PUFAs (total)	-	
	EPA-DHA	DHA 100 mg/day from 7 months	
Children and adolescents (1–17 years)	PUFAs	-	
	ω-6 PUFAs	-	
	ω-3 PUFAs (total)	Total: - ALA: 0.5% En (from 2 years)	
	EPA-DHA	DHA 100 mg/day until 2 years EPA + DHA 250 mg/day from 2 years	
Pregnant and lactating women	PUFAs	-	
	ω-6 PUFAs	-	
	ω-3 PUFAs (total)	Total: - ALA: 0.5% En	
			EPA + DHA 250 mg/day + DHA 100–200 mg/day

Abbreviations: AI: Adequate Intake. RI: Reference Intake. En: total energy.

**Table 3 nutrients-14-00152-t003:** Clinical evidence on ω-3 PUFAs supplementation and the occurrence of FA or food allergens sensitization.

Reference	Subjects and Number	Supplementation	Time	Outcome	Results
Dunstan JA, et al. J Allergy Clin Immunol. 2003 [75]	40 atopic pregnant women	Fish oil group: 4 (1-g) fish oil capsules per day comprising a total of 3.7 g of ω-3 PUFAs with 56.0% as DHA and 27.7% as EPA. Control group: 4 (1-g) capsules of olive oil per day (containing 66.6% n-9 oleic acid and <1% ω-3 PUFAs)	From 20 weeks gestation until delivery	Neonatal PUFAs levels and immunologic response to allergens at birth and clinical evaluation at 12 months of age as a secondary outcome	All neonatal cytokine responses to all allergens tended to be lower in the fish oil group; infants in the fish oil group were three times less likely to be sensitized to egg allergen at 1 year of age
Furuhjelm C, et al. Acta Paediatr. 2009 [68]	145 pregnant women, affected by allergy themselves or having a husband or previous child with allergies	Daily supplementation with either 1.6 g EPA and 1.1 g DHA or placebo	From the 25(th) gestational week to average 3–4 months of breastfeeding	The incidence of allergic disease in the first year of infants’ life	Reduction of the FA risk and IgE-associated eczema during the first year of life
Manley BJ, et al. Pediatrics. 2011 [76]	657 breastfed preterm infants (<33 weeks’ gestation)	Mothers taking either tuna oil (high-DHA diet, ∼1% total fatty acids) or soy oil (standard-DHA, ∼0.3% total fatty acids) capsules	From 2–4 days of postnatal age until 40 weeks postmenstrual age	Incidence of bronchopulmonary dysplasia and parental reporting of atopic conditions over the first 18 months of life	No effect on the incidence of parental reported FA
Furuhjelm C, et al. PediatrAllergyImmunol. 2011 [77]	145 pregnant women, affected by allergy themselves or having a husband or previous child with allergies	Daily supplementation with either 1.6 g EPA and 1.1 g DHA or placebo	From the 25(th) gestational week to average 3–4 months of breastfeeding	IgE-associated diseases last up to 2 years of age and assess the relationship between plasma proportions of ω-3 PUFAs and the frequency and severity of infant allergic disease	Decrease in cumulative incidence of IgE-associated disease; high proportions of DHA and EPA in maternal and infant plasma phospholipids were associated with less IgE-associated disease and a reduced severity of the allergic phenotype
Palmer DJ, et al. BMJ. 2012 [69]	706 pregnant women with a fetus at high risk of allergic disease	Fish oil capsules (providing 900 mg of ω-3 PUFAs daily) or matched vegetable oil capsules without ω-3 PUFAs (control group)	From 21 weeks gestation until delivery	IgE associated allergic disease (eczema or FA with sensitisation) at 1 year of age	No reduction in the overall incidence of IgE associated allergies; lower atopic eczema and egg sensitisation
D’Vaz N, et al. Pediatrics. 2012 [78]	420 infants at high atopic risk	Daily supplement of fish oil containing 280 mg DHA and 110 mg EPA or a control (olive oil)	From birth to age 6 months	Occurrence of allergic outcomes including sensitization, eczema, asthma, or FA	No effects in the prevention of childhood allergic disease
Best KP, et al. World Allergy Organ J. 2018 [79]	706 pregnant women with a fetus at high risk of allergic disease	Fish oil capsules (providing 900 mg of ω-3 PUFAs daily) or matched vegetable oil capsules without ω-3 PUFAs (control group)	From 21 weeks gestation until delivery	IgE associated allergic disease (eczema or FA with sensitisation) at 3 and 6 years of age	No significant effects on IgE-mediated allergic disease symptoms or sensitization

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
