# Peer review of "Potential Role of Omega-3 Polyunsaturated Fatty Acids in Pediatric Food Allergy"

_nutrients, 2021, doi:10.3390/nu14010152_

Round 1
Reviewer 1 Report
The review that Sartorio et al. have submitted describes the potential role of n-3 PUFAs in pediatric food allergy (FA). The review is well constructed and referenced, was enjoyable to read and provides an exhaustive overview of this particular field of interest and research. There are several areas of concern to be addressed however:
Most importantly, while the title indicates that the review is focused on pediatric FA, a small but significant amount of text is dedicated to other allergic diseases/responses:
-
- In section 6. "PUFAs supplementation for food allergy prevention:preclinical findings", Lines 160-167 discuss allergy models which are distinct from FA
- Similarly, section 8 "PUFAs supplementation in infancy and childhood for allergy prevention" dedicates a paragraph to asthma and wheezing.
While this is not necessarily a problem, since allergic diseases are somewhat connected, it requires more context, so that the reader does not get lost and assumes that n-3 PUFA will exert similar activities in different settings, which is highly unlikely. As the authors point out, "It is necessary to intensify the scientific efforts in order to clarify their actual effects", and since these effects will be context specific, the authors need to point out when they reference work which does not strictly apply to FA. It may also be beneficial to introduce the concept of allergic march in this review, which may clarify for the reader why anatomically distant allergic responses are being discussed, and why a skin- or lung-associated n-3 PUFA effect may have relevance for FA.
I would also tone down the statement made in lines 111-112, that the "EAACI failed to demonstrate a protective role of breastfeeding in allergy prevention", since that observation had a low certainty.
In the paragraph 174-190, I missed the perspective that EPA/DHA more generally reduce the immunogenicity of DCs, not just their ability to suppress Th2 responses. This has to be clarified for the benefit of the reader. Costimulatory molecules such as CD80/86 also participate in Th1 responses and IL-12p40/IL-23 are drivers of Th1/Th17 responses.
Minor comments:
- n-3 or ω-3: please stick to one instead of mixing both throughout the text
- Th2 or Th-2: again, choose one
Reviewer 2 Report
1.Diet Diversity was described in paper but this often relates to fibre and plant diversity and non-processed foods is LCPUFA intake considered to be a isolated problem or simply that having high LCPUFA as part of diverse diet is the reason why data can be so variable in different regions and population
2.,Countries where Vegetarian is more common would you expect allergy to be higher and rise of Vegan in subpopulations in high economic incomes is their any commentary of the possible rise of allergy /increased risk in these populations or subpopulations
3. Neonatal dosing 600mg-1000mg early for preterm prevention's an increasing area on interests would that pro-inflammtory of reduce FA risk or is the data unknown - Ie By the Neonatal specialist drive to use high dose may become common practice to reduce pre-term births
4. Non-IgE mediated allergy is not mentioned in paper is there any data or comments to add into this type of allergy
Reviewer 3 Report
The protective effect of omega-3 polyunsaturated fatty acids (n-3 PUFA) on inflammation and allergic response have been reported in the literature. The positive feature of this paper is the extensive summary on how n-3 PUFA can potentially be used to improve food allergy prevalence in pediatric population. The lack of authors’ interpretation and analyses of the findings from various studies reported in the review is a drawback to the paper. Overall, the review offers an interesting perspective on potential use of n-3 PUFA supplementation to improve food allergy in children.
Major points
- In Section 6, p. 5, paragraph 2 – 4, the authors state that polymorphism of FADS gene cluster leads to reduced PUFA serum levels. The authors should consider reporting the prevalence of FADS polymorphism, specifically in pediatric population.
Minor points
- The abbreviations used should be defined when they first appear in the text, e.g. p. 2, line 47, Th2 should be defined as T helper cell 2 (Th2); p.4, line 138, sIgE should be defined as specific IgE.
- The abbreviations used should be consistent throughout the manuscript. When the authors refer to omega-3 PUFAs, they should stick to one abbreviation throughout the text. Moreover, when the authors refer to long chain polyunsaturated fatty acids, they should stick to one abbreviation (LC-PUFA or LCPUFA).
- On p. 5, line 191, there should be an indentation to indicate a new paragraph. The authors might also want to consider merging this paragraph with the two paragraphs that follow this one.
- On p. 7, line 153, the word “or not” should be omitted from the sentence.
- On p. 7, line 268, there is a mistype on meta-analysis.
- On p. 8, line 296, antioxidant should be one word without a hyphen.
Round 2
Reviewer 1 Report
The authors have addressed all of my comments.